# Statelessness and mental health experiences of Kuwaiti Bidoon people living in the UK: An interpretative phenomenological analysis

Sana Zard[1], Ciarán O'Driscoll[1], Jessie Mulcaire[1], Leah Holt[1], Sasha Menon[2], Francesca Brady[1,3]*

1 Research Department of Clinical, Educational and Health Psychology, University College London, London, United Kingdom, 2 Department of Psychiatry, University of Oxford, United Kingdom, 3 Helen Bamber Foundation, London, United Kingdom

* f.brady@ucl.ac.uk

## Abstract

The Kuwaiti Bidoon are a group of people affected by statelessness. Estimates suggest thousands of Kuwaiti Bidoon have forcibly migrated to the United Kingdom (UK); however, little is known about their experiences of mental health. This study aimed to explore the mental health experiences of statelessness among Kuwaiti Bidoon people living in the UK, and their experiences of accessing mental health services (where indicated).Participants were five Kuwaiti Bidoon people currently living in the UK. All participants attended a semi-structured interview. Experiences relating to statelessness and mental health were investigated using Interpretative Phenomenological Analysis. Participants shared the multifaceted impacts of statelessness on their lives, including mental health struggles stemming from their marginalisation and uncertain legal status. Three major themes were generated from the interview data: The Legacy of Statelessness; Hopes and Dreams of a Future; Victims of a System. Hope and optimism arise for some when migrating to the UK, while others reported challenges and distress associated with the state of 'limbo' arising from processes to regularise their legal status. Some participants reported barriers to accessing effective mental health support, which was sometimes connected to their legal status. This study raises awareness of the context for UK-based Bidoon people and furthers understanding of the long-term negative mental health consequences of statelessness. Further research directions, recommendations for improvements to healthcare and statutory service provision for stateless or displaced people (such as ensuring accessibility, acceptability and delivery of culturally sensitive care), and the need for broader policy change are discussed.

## Introduction

The Universal Declaration of Human Rights (UDHR) asserts every individual has the right to a nationality [1]. However, up to 15 million people globally are affected by

**Data availability statement:** The data for this article cannot be shared in an unrestricted way due to the privacy concerns of the participants included this study. The consent form approved by the University College London Ethics Review Board did not cover the sharing of data beyond the research team (or as aggregated in publications); this decision was based on strong recommendations from community consultants. To request data underlying this manuscript, please contact the corresponding author. You may also contact the University College London Ethics Review Board for any questions or access queries at ethics@ucl.ac.uk (ref: 21409/001).

**Funding:** The authors received no specific funding for this work.

**Competing interests:** The authors have declared that no competing interests exist.

statelessness [2], i.e., they are "not considered as a national by any State under the operation of its law" (United Nations Convention relating to the Status of Stateless Individuals) [1]. Statelessness has wide ranging consequences. This can range from restricted access to identity documents, to being denied basic rights including to education, healthcare, employment, housing, and freedom of movement [3]. Many live in extreme marginalisation, facing exploitation, violence, and persecution, often resulting from the intersection of statelessness and minority group status [4,5]. Stateless people encounter a wide range of risk factors that may negatively impact their mental health and often lack protective factors that can strengthen psychological well-being, such as community inclusion, education and employment opportunities, and access to healthcare [6,7].

The Kuwaiti Bidoon are a group of stateless people with an estimated population between 88,000 and 106,000 [8]. The Arabic phrase *Bidoon Jinsiya* can be translated to 'without nationality' in English and is often shortened to Bidoon, Bedun, Bidoun or Bidun [9]. This report uses the term 'Bidoon' to denote the Kuwaiti Bidoon specifically, although there are other stateless groups across the Middle East who are also known colloquially by the term Bidoon.

Kuwaiti Bidoon statelessness originates from Kuwait gaining independence in 1961 and requiring citizens to register [10]. Over a third of residents at the time (many who came from tribal backgrounds) did not register for citizenship, either because of a lack of awareness or adequate documentation [11]. Subsequent amendments to Kuwaiti nationality laws further restricted Bidoon community access to citizenship [10].

Since 1985, the Kuwaiti government have classified Bidoon as 'illegal residents' and perpetuated a narrative of them as hostile foreign nationals, which has fostered discrimination and hostility against the community [11,12]. The legal and administrative status as 'illegal residents' in Kuwait exposes Bidoon to structural, systemic, and administrative violence [12]. While Kuwaiti citizens receive free healthcare, education, and subsidies, Bidoon are largely excluded [12]. Bidoon are required to carry a valid security card, which can be used to access some services, however, these do not match the entitlements available to citizens and there are reports that provision is inadequate to meet basic healthcare needs [8]. Similar restrictions are placed on legal employment opportunities. Bidoon can pay for private education if they are able to afford this, however, the private education system is reportedly of a lower standard than government schools. Nevertheless, many have no income to afford private schooling, therefore contributing to low educational attainment in the community [12]. Thus, Bidoon often work in low-paid, informal employment and live in deprived neighbourhoods [12,13].

Most academic literature on the Bidoon comes from anthropology, sociology, law, and political disciplines [3]. Like other stateless groups, Bidoon face multiple structural and social risk factors associated with poor mental health outcomes, but comprehensive research exploring mental health in this community is limited. However, the available literature indicates connections between Bidoons' experience of statelessness and poor mental health and wellbeing. For example, Bidoon students

at Kuwait University reported that their education (and career) options were limited by their legal status, which led to a deterioration in mood [14]. Another study found that Bidoon adolescents who had experienced family disruption or relocation were more likely to report suicidal ideation [15]. This was mediated by a lack of attachment to norms within their local community, suggesting that social disconnection may have a detrimental impact on mental health.

Political alienation among the Bidoon was explored in a survey which found high levels of powerlessness, isolation, and normlessness [16], though the mental health impact of this remains unexplored. As a result of the alienation and persecution experienced in Kuwait, some Bidoon are forced to migrate [17]. In the UK, Bidoon may seek protection through the asylum system or the Statelessness Determination Procedure (SDP) [18]. However, both applications routes require a high burden of proof [19,20] and are prone to lengthy decision times [20]. One media estimate suggests around 5000–8000 Bidoon people reside in the UK [21], but no official or precise Home Office data exists regarding exact numbers or current immigration status (European Network on Statelessness [19]). According to data collected by ENS, the exact number of stateless people in the UK has proven very hard to determine, largely due to stateless individuals fearing repercussions from their home country and the Home Office if they declare their status [22]. This lack of accurate data makes it challenging to understand the physical and mental health needs of stateless people in general, let alone specific communities such as the Kuwait Bidoon. This, in turn, makes it difficult to establish the types of healthcare interventions that might be appropriate for such communities or individuals.

Due to their marginalisation in both Kuwait and the UK, little is known about the mental health of Bidoon people. In the absence of comprehensive research exploring mental health of Bidoon people in the UK, broader research with other forcibly displaced groups may offer some insight [23]. One participatory assessment conducted by the UN with 12 stateless people undergoing the SDP included two Bidoon [3]. Participants who were awaiting a decision from the SDP connected the length of time they had waited for a decision along with the experiences of going through the legal process with poor mental health, with some participants reporting feelings of hopelessness, depression, and suicidal ideation. These difficulties are likely further compounded by Home Office regulations barring stateless people awaiting a decision on their SDP application from having the right to work, accessing public funds, obtaining social housing, free secondary healthcare under the UK's NHS, and legal aid funds [24,25]. A multi-non-governmental organisation briefing on stateless people in the UK found that the lengthy asylum and SDP application timelines, when combined with the barriers of being able to work or access essential services may leave stateless people in 'legal limbo' and at a heightened risk of destitution and exploitation [26].

Research with asylum seekers and refugees also report higher incidence rates of depression, post-traumatic stress disorder (PTSD), and anxiety disorders, compared to the general population [27] and pre-displacement factors (including trauma and experiences of persecution) that can impact negatively on mental health [28]. Additionally, post-migratory and integration challenges in host countries have also been associated with poorer psychological wellbeing, including housing and financial difficulties, unemployment, social isolation, family separation, and difficulties navigating the asylum process [29,30]. Stateless people also face additional barriers to providing sufficient evidence to the Home Office [3]. This can result in long waits to receive an asylum decision and rejected claims which have been linked to poor mental health outcomes [31–33].

Currently, there is no research into the mental health experiences of Bidoon people in the UK and how these relate to their experiences of statelessness or their perspectives about accessing support or mental health services in the UK. This research seeks to fill this gap with a view to informing clinical professionals about the impact of statelessness on mental health for Bidoon people, whilst also highlighting more general implications for adapting interventions and service provision for stateless people.

The research questions for this study are:

1) What are the experiences of statelessness and mental health among Kuwaiti Bidoon people living in the UK?

2) What are their experiences of accessing support or mental health services (where indicated) within the UK?

PLOS Mental Health

## Materials and methods

### Participants

Connections with community partners and other organisations, including charities, researchers and non-governmental organisations who work with Bidoon people in the UK were established in the design phase of the project. Partner organisations were briefed on the aims and eligibility criteria and invited potential participants to take part in the study. IPA requires shared experience among participants, therefore lending itself to a purposive sampling approach [34]. This is a non-probability sampling method that is employed when it is necessary to recruit participants with characteristics that meet the aims of the study.

Participants were eligible to take part in the study if they 1) identified as Bidoon, 2) were born in Kuwait, 3) currently reside in the UK, and 4) were aged 18 or over. Individuals with any type of UK immigration status were eligible, and there was no requirement with regards to the length of time they had resided in the UK. Both English and non-English speakers were included.

The recommended sample size for IPA is 4–10 participants [34]. Eight people were originally recruited to the study, however, three withdrew consent —two over concerns about anonymity (despite reassurances about how data would be used and stored), and one did not attend the interview or respond to further contact. Ultimately, five participants (four men, one woman) took part, aged between 20 and 60 (mean age = 33). Specific ages, demographic characteristics and recruitment details of the participants have been omitted to protect anonymity (given the small size of the UK community). Participants held different immigration statuses: one British citizen (former refugee), two current asylum seekers, and two with refused claims. Two interviews were conducted in English; three required an Arabic interpreter.

### Ethics statement

This study was approved by the Ethics Committee of the University College London (ref: 21409/001) and was conducted in accordance with the ethical principles outlined in the Declaration of Helsinki. Full informed consent was obtained. Consent procedures are outlined in the Procedure section below.

### Procedure

Members of the UK-based Bidoon community, and those working closely with the community were consulted across four community engagement meetings to ensure their voices shaped the design of the research project. Community partners advised on the information sheet, interview schedule, and effective recruitment methods. Recruitment took place between 18 July 2022 and 01 June 2023.

Participants were recruited through community organisations and networks (some networks or organisations are quite small, and therefore are not named here to protect the identity of the participants). The study was advertised by a flyer, through word-of-mouth, and via social media applications such as WhatsApp by the community organisations. Interested participants contacted the researcher directly, or were initially provided with information about the study by the community organisation. English speakers received a written information sheet in English, whereas Arabic speakers received a video translation and a written Arabic translation of the information sheet. Informed written consent was obtained (with translation as required) prior to participation. Where literacy was a barrier, recorded oral consent was gathered instead (with the assistance of an Arabic interpreter as required). Participants were offered a choice between in-person interviews, or virtual interviews using Microsoft Teams. All participants opted for a virtual interview.

Only minimal demographic information (age, gender and immigration status) was collected from participants, which was based on recommendations from community consultations; they had suggested that requiring detailed personal information to be shared may reduce willingness to participate in the study. Interviews were semi-structured using an interview schedule that included broad questions drawn from relevant literature and clinical experience. It was developed in

collaboration with community partners who made recommendations about content and language. Interviews were initially required to be audio recorded, but after concerns were raised by some potential participants (see Participants section), those taking part were also offered the option of their responses being fully transcribed rather than recorded. Unfortunately, two participants still declined to be included as a result of concerns about anonymity and use of their data. Those that did participate agreed to audio recording; the recording was subsequently deleted upon transcription of the interview.

The same Arabic interpreter was used where participants required it to ensure consistency. The interpreter was professionally trained and worked regularly in an NHS service set up to support forcibly displaced people (which includes some Bidoon individuals). Further, they were provided specific training by the study authors about the aims of this study to facilitate effective and comprehensive interpretation. The interpreter translated questions and responses in real time, and the English translation was transcribed for analysis. A briefing before each interview and a debrief afterwards helped minimise any loss of meaning and cultural nuance.

Following interview, participants were offered a debrief and provided with information about relevant support services as required. Participants were reimbursed for their time with a 'High Street' electronic gift voucher worth £20 which could be redeemed at a variety of retail stores based on the participant's choice.

## Data and analysis

Whilst all possible steps have been taken to protect the identity of the participants in this study, given the small size of the community in the UK and the valid concerns that participants hold about the risks if they were able to be identified, interview transcript data had not been made available in an unrestricted way. Interpretative Phenomenological Analysis (IPA; [34]) was chosen to facilitate in-depth exploration of the lived experiences and meaning made by participants about these experiences. IPA is rooted within each participants' personal experience as well as highlighting convergent and divergent themes across participants. The interpretation is two-fold based on how participants make sense of their experience, and how the researcher makes sense of this [34]. Interviews were transcribed and then analysed followed the six stages set out by Smith [34]: 1) reading and rereading; 2) initial coding; 3) developing emergent themes; 4) searching for connections across emergent themes; 5) moving to the next case; and 6) looking for patterns across cases.

## Reflexive statement

All authors of this paper have professional experience working with asylum seekers and refugees in the NHS. The first author undertook a bracketing interview [35] to explore their assumptions and documented reflexive insights in a journal throughout the research process [36].

## Results

Three major themes and 14 sub-themes were generated, as outlined in Table 1 below.

## 1. The legacy of statelessness

All participants contributed to this major theme, which captures how statelessness continues to influence the lives of participants. Struggles are rooted in past experiences and the way these have shaped their sense of themselves, their difficulties, and place in the world.

**1.1. Sense of Self.** All but one participant described how growing up Bidoon shaped their identity. Participant Two described being deprived of human rights in the form of access to basic services led to their sense of self as non-existent: "*We don't exist. Uh, we're not allowed to have any rights in education, work, medical care…*"

Participant Three labelled themselves a "*second-hand person*" in comparison to a Kuwaiti citizen who has access to rights and services unavailable to the Bidoon community: "*We were marginalized. We knew that we were like you know, second hand people…*"

**Table 1. Themes and Sub-Themes.**

| Major Theme | Sub-Theme |
| --- | --- |
| 1. The Legacy of Statelessness | 1.1 Sense of Self<br>1.2 Mental and Emotional Strain<br>1.3 Existing Rather than Living<br>1.4 "Scattered Selves"<br>1.5 Silenced |
| 2. Hopes and Dreams of a Future | 2.1 Changing Location, Changing Perspective<br>2.2 Empowered by a Cause<br>2.3 A failed promise<br>2.3.1 "Two Hits on the Head is Very Painful"<br>2.3.2 State of Limbo |
| 3. Victims of a System | 3.1 Barriers to Accessing Mental Healthcare<br>3.2 Painful Discriminatory Experiences<br>3.3 Belonging and Connection Despite Discrimination<br>3.4 Beyond the Individual |

**1.2. Mental and emotional strain.** Most participants shared that their experience of statelessness was associated with current distress. For one participant there was a dominant narrative of past threats intruding into their present life. It seemed this was related to their past traumatic experiences of imprisonment.

*"I left Kuwait and from that time until that moment I had problems. I was sometimes my [spouse] would tell me, you know, I shout a lot when I'm sleeping…You know, but sometimes I feel...someone is chasing me, you know, sometimes I hear I hear strikes and I hear like cries…"* (Participant One)

One used the concept of *"depression"* to make sense of their experience of feeling powerless and hopeless.

*"It left me with depression…which is, I'm aware, a feeling that is completely different to sadness. Depression to me, mean that the problem is I decided to fight and to use all my powers, my energy, all I cannot accept …The problem is every time I try to do something to change the situation, I end up on my own….nothing is coming out of those attempts."* (Participant Two)

Another participant was consumed by distressing thoughts about the predicament of family members in Kuwait.

*"When I spend a long time on my own, that means I will start to think a lot, miss my family a lot, to feel upset that I'm away from my family and that they're not in a good situation."* (Participant 5)

**1.3. Existing rather than living.** This theme describes the experience of two participants, both of whom were refused asylum seekers. They shared a sense of disengagement with their life. One participant described a state of powerlessness to change their situation due to past failed attempts.

*"…the feeling I have at the moment is very uh, very weird because I know that I can't change anything. It's something out of my power out of my control and nothing I can do with change their minds or change the way they treat us."* (Participant Two)

For another participant, any desire for a better life has been extinguished due to the ongoing impact of early life experiences of deprivation.

*"I lost all kinds of motivations and all kinds of desires. Uh, to live a meaningful life. I'm 45 years old now. I was deprived of education. Away from my family, I'm still not married. I don't have children. I don't have, uh that the capacity to have a good job here. I don't know the language."* (Participant Four)

**1.4. "Scattered selves".** Most participants described loss and longing for who or what they had left behind in Kuwait. Immigration laws in both Kuwait and the UK exacerbated this sense of loss. Participant One used the powerful metaphor of a *"scattered self"*. They understood themselves as incomplete unless united with their family.

*"I think I'm scattered...all the time I'm thinking about my father that even if my case is accepted by the Home Office, even if my [spouse and children] came here. I still, part of me in Kuwait because of my mother and sisters, who, impossible to come, all of them here."* (Participant One)

One participant spoke of a longing for the culture and religion left behind; they expressed feeling that their values were better represented by Kuwaiti culture and that adjustment to British culture and customs had been challenging.

*"The thing that I favoured about back home in Kuwait is that I was surrounded by a more Islamic and conservative community. So I felt more comfortable around that community, especially that I knew I'm in my country and with my family…"* (Participant Five)

**1.5. Silenced.** Two participants spoke of being silenced and not being able to speak freely due to a sense of being controlled by the Kuwaiti government. Participant One spoke of the need to censor themselves out of a fear for the safety of their family: *"Whatever you do here affects your family there, so you need to be careful."*

Another participant felt the same; even though they now have regularised status in the UK, they did not feel able to engage with their right to express themselves freely (as other British citizens might) because they continue to worry about repercussions towards their family in Kuwait should they talk freely about the plight of the Bidoon.

*"It's very stressful. You know, I, even if you become a British person, OK, then you say, OK, everything we pass. I'm not stateless anymore. But why am I still being controlled, and I can't express, I can't even talk about my cause."* (Participant Three)

## 2. Hopes and dreams of a future

All participants contributed to this theme, describing a sense of hope that had accompanied their move to the UK. Hope empowered participants to seek a different life in the UK and to keep fighting for change. When hope was dashed, often due to refused asylum claims, the impact was devasting.

**2.1. Changing location, changing perspective.** Three participants experienced a shift in perspective associated with their move to the UK. One spoke about how therapy had facilitated a shift in their self-narrative from victim to survivor.

*"…in the beginning [of therapy], I was like kind of, you know externalizing what I have internalized in Kuwait. So the first two sessions, I was kind of showing myself as someone who was like, victimized, my CBT [therapist] was saying so what you're strong person? No, no. Like it was like very empowering, I think."* (Participant One)

For another participant, coming to the UK helped them feel hopeful and 'unstuck' from their previous position as a person with a lack of agency. Their story contained a contrast between a life with few prospects in Kuwait to one that is more hopeful and purposeful in the UK.

*"… I found like my life has changed and now I can have aspirations and hopes, and I became hopeful…I now can have hopes and aspirations always happy. I like to go outside. I like to join gatherings and be around people, especially to learn from them and learn new things…I have to explore options of jobs I can do."* (Participant Five)

**2.2. Empowered by a cause.** For three participants, fighting for their community's rights was empowering and offered a sense of hope. One expressed a need to speak truth to power that underpinned their decision to challenge the discrimination they had faced from the Kuwaiti government.

*"… I am not, you know, inciting violence. Nothing. I'm just, I have very constructive points of view, if you like to listen to them, OK? If you don't, it's OK also, but don't interfere with, but don't mix up, you know my private life with my political expressions. So this is what I want to challenge… I will never give up on my project, you know to continue speaking up because really there is a big job waiting…"* (Participant One)

Another participant was empowered to help others in the community; they felt a duty to help others and a sense of optimism that they can be the change they want to see in the community.

*"This is a personal experience. I don't see other people empowered to do volunteer work. This is totally personal. For me. And simply as a person who wants positivity. That's how I was empowered. I want to do something good for these people. I see these people are struggling […] And that for me, was more than enough to have some social responsibility and being empowered."* (Participant Three)

**2.3. A failed promise.** This sub-theme captures the expectations, reality, and experiences of moving to the UK. There are two parts: '"Two hits on the head is very painful"' and 'state of limbo'.
***"Two Hits on the Head is Very Painful".*** Two participants whose asylum or protection claims had been refused contributed towards this theme, which describes the cumulative impact of disappointment. One participant spoke of the trauma of growing up as Bidoon in Kuwait, as well as the trauma of daring to dream for a different life in the UK which did not materialise. They shared painful experiences of marginalisation and alienation in both Kuwait and the UK. They gave a powerful metaphor to describe this situation.

*"I thought that I will go somewhere else. Where people, where people see me as a human and treat me as a human… Back home in Kuwait. Now here in the UK. We have a saying in Arabic, it's two hits on the head is very painful. And now I had the two hits, one back home and one in the UK."* (Participant Two)

Another participant described a palpable sense of disappointment and hopelessness after striving for a better life and being thwarted by the UK asylum system.

*"… in Kuwait, yes, I was struggling, but I had hope that one day I will leave this country and maybe start again. But when I came to this country hoping that I will build up a new life, new future and they declined me, they ruined everything, and I became hopeless."* (Participant Four)

***State of Limbo.*** Three participants described their experience of the UK asylum system as a state of limbo. One participant described feeling stuck between denial of human rights in Kuwait and refusal of asylum in the UK.

*"Umm things are getting like overwhelming and suffocating more and more over the time I feel like I'm in the middle of the river, I can't jump to the ground and I can't continue."* (Participant Two)

For another participant this feeling of being in limbo was manifested through a lack of urgency on the behalf of statutory services to access healthcare.

*"I'm very upset and scared because I have a problem, an illness and I need the treatment for this problem…I want my status to be able to work and money and then have this treatment."* (Participant Five)

### 3. Victims of a system

All participants contributed to this major theme, which captured the experiences of navigating statutory services in the UK (such as healthcare, immigration, and housing). Experiences were shared of systemic and structural failings, but also of genuine human connection that fostered positive wellbeing. Participants reported feeling that mental health care services are not equipped to address structural causes of poor mental health and sometimes could do more harm than good.

**3.1. Barriers to accessing mental healthcare.**  Four participants experienced barriers to accessing mental healthcare. Participant Five expressed a lack of knowledge about how to access and navigate support services in the UK.

For another participant, there was a sense that the support they received came too late. Underpinning this was a feeling that support services do not fully understand the plight or experiences of those seeking protection in the UK. They expressed frustration about the rigidity of services in the UK.

*"…I started to have mental health care 18 month after my [asylum] interview. I had my interview one month after arriving in the UK, which was pointless because yeah, that interview was the most important thing for me as a refugee or asylum seeker, and I was wishing to be helped beforehand, not after."* (Participant Four)

In their role supporting those in the community, Participant Three undertook a Mental Health First Aid course which shaped their perspective of mental health. They shared that others in the community may be discouraged from seeking support from mental health services because of cultural notions that mental health treatment is shameful: *"…it's shameful basically. They say it's shameful, I don't know why, that's from the people that I know. That's shameful, they see it as shameful. Shameful."*

**3.2. Painful discriminatory experiences.**  Two participants spoke of experiencing discrimination when accessing statutory services in the UK. One participant explained that feeling rejected by health services had led to their decision to reject health services. Such rejection by healthcare services may also mirror experiences of discrimination faced in Kuwait, exacerbating the impact of this experience.

*"…I said I'm drowning in my blood. I, I understand English but not to the level that I can express myself fully all the time. Umm. And when I said that he laugh, he said then you can hold on to the ceiling so you don't drown. Umm. And they laughed. And when I said I'm sorry I called the ambulance. He didn't like the way I pronounced the word ambulance, so he made fun of this as well. So why would I share my feelings with someone?"* (Participant Two)

Another participant described experiencing racism from the UK's Home Office, which had a lasting impact.

*"I thought that my life is one hundred per cent destroyed. Specially because of the way I was treated by the Home Office, their stubbornness and their cruelty, and part of it, I think, was racism as well. I can remember all the details about that. A member of staff in the Home Office who was very, very racist and the way he talked to me how he looked, the, the insults and how he treated me are the still stuck in my head and I still remember all the, those details very well until this moment."* (Participant Four)

**3.3. Belonging and connection despite discrimination.** Most participants contributed towards this theme, which described experiences of belonging and connection despite experiencing discrimination. Being recognised as a human, as an equal, and having genuine and meaningful connection with others can provide an antidote to the systemic dehumanisation perpetrated by governments.

One participant spoke of being reminded of their essential humanity through engagement with healthcare services.

*"So as I told you, mental health. First of all, gave me a kind of reassurance that I am a human being. Uh, because I was deprived from healthcare in Kuwait and when I see someone in the UK offering me their help, you know this is a good feeling, you know? That I'm not alone. I am not abandoned…"* (Participant One)

Another participant spoke of a strong desire to feel heard by another human, especially when feeling isolated following their move to the UK. They found therapy helpful, in contrast to feeling uncared for by the wider healthcare and immigration systems. Individual connection informed their recovery and perceptions of their situation.

*"The mental health support that I received was to find someone to talk to in the session. Yes, it was a temporary and small and limited improvement, but it was essential for me at that point of time. That I'm now having support and someone that cares…When I talk to them, I feel like I'm talking to my friend and that this is someone who is listening to me, who's offering the space for me to share my feelings, and they're offering their sympathy."* (Participant Four)

**3.4 Beyond the individual.** Two participants spoke of their experiences of finding mental health services to be ineffective when they sought help. They expressed that healthcare services and professionals should do more to challenge the wider systems that are failing to support or protect them effectively, and the tackle the societal injustice that the Bidoon experience.

One participant shared an experience of feeling suicidal and calling a suicide support service. They expressed frustration at limitations in the system and the lack of joined-up care across agencies. They felt let down and discouraged from help-seeking.

*"…the least I would expect is the person would call the police and said, excuse me, try to help this person. Try to call the authorities. Try to call the Home Office call anyone. Because this is a human being's life. But when they say sorry, we can't do anything with your asylum, we can't do anything with your paperwork…. That discouraged me, actually."* (Participant One)

Another participant spoke of the social and structural inequalities and struggles that other members of their community face. Implied within their response was that mental health should not be located solely within the individual, but conceptualised and addressed within the wider socio-political context that may have led to their distress.

*"Most of the time they have a psychological problem. And that's in the UK, I'm not talking about, uh, Kuwait… I think it's a sad feeling when you feel left out, simply. You know, as a stateless person in the UK. And most of us, let's say the parents' generation, they lack a proper language. They lack experiences. They don't know enough. That's why a lot of them, they prefer to go to, to stay on the benefits and do bad work, like low wages."* (Participant Three)

## Discussion

This study explored the experiences of statelessness and mental health among Kuwaiti Bidoon people living in the UK, including experiences of accessing mental health services. Five individuals were interviewed and data was analysed using IPA. Three major themes were generated: The Legacy of Statelessness, Hopes and Dreams of a Future, and Victims of a System.

### The legacy of statelessness

All participants shared experiences about the myriad of ways statelessness had impacted their mental and emotional wellbeing, regardless of their current legal status. This is compounded by post-displacement factors experienced in the UK, such as separation from family members in Kuwait. In Kuwait, participants faced systemic denial of rights (citizenship, healthcare, education, and employment) leading to significant social and economic exclusion. These conditions align with risk factors in the social determinants of mental health [6] and have implications for understanding the needs of not only the Kuwaiti Bidoon, but also other displaced or stateless groups or individuals [5].

Early exposure to deprivation due to their stateless status appeared to have a formative impact on participants' sense of self as adults. These impacts can be understood through the lens of adverse childhood experiences (ACEs) [37], which are linked to long-term physical and mental health outcomes in later life [38]. As a result of social and economic disadvantages, many from the Bidoon community (as well as other communities or individuals experiencing statelessness), may have been exposed to multiple ACEs [37,39]. For example, daily stressors associated with statelessness may have led to untreated mental health difficulties in parents, with downstream effects on their children. It is therefore important to consider the prospect of intergenerational trauma, where trauma-related symptoms are transmitted across generations [40]. Thus, it may not be solely the stateless legal status that is passed on to children, but also the emotional traumatic legacy of parental experiences of statelessness. Further research exploring intergenerational trauma within the Bidoon community, and across other communities and families affected by statelessness is required to explore this issue further.

### Hopes and dreams of a future

For some participants, moving to the UK facilitated a sense of hope and optimism about their situation and prospects. Broader research suggests hope can predict wellbeing [41], and may therefore be an important factor to consider in the wellbeing of stateless or displaced people more broadly. However, uncertainty related to a person's asylum claim or legal status undermined this hope. The sub-theme 'State of Limbo' encapsulated the anxiety experienced by both current and refused asylum seekers who were awaiting decisions on their immigration matters. Fears of being returned to unsafe conditions in Kuwait, combined with a lack of legal documentation or certainty, created a prolonged state of insecurity and vulnerability. This has long been recognised as a significant and profoundly existential concern for those seeking asylum or other forms of regularised status in a host country [42]. Participants who had been refused asylum could not return to Kuwait or travel elsewhere, leaving them at risk of detention or destitution in the UK. The one participant granted citizenship did not contribute to this theme, perhaps highlighting the protective role of obtaining a secure immigration status.

Participants also noted the health consequences of delayed asylum (or other protection applications) decisions. Wider research links longer immigration decision waiting times to poorer health [33,43]. Moreover, for those who had been refused protection in the UK, dashed expectations resulted in acute disappointment and hopelessness. Studies confirm that hopelessness is associated with depression and suicidal ideation [44,45] and rejected asylum seekers show worse mental health outcomes [32], and are less likely to access support, despite being at heightened risk of developing mental health difficulties [46]. These findings therefore further underline the need for prompt decision-making for legal protection claims and suitable healthcare that is accessible throughout this process.

### Victims of a system

This theme captured participants' experiences of and interactions with UK support services at the structural and individual levels. Some participants encountered barriers to accessing mental healthcare that were structural in nature, including delays to accessing treatment due to waiting times. One participant reported lacking information about how to even access services. Another described how cultural notions of mental health and stigma associated with it may deter others

in the community from accessing support. These concerns are not limited to the Kuwaiti Bidoon community; these findings mirror those reported by refugee and other forcibly displaced individuals, [23,30,47], which often includes perceived barriers specifically pertinent to the individual's migration or legal status. This suggests longstanding and systemic barriers, as well as potentially discriminatory practice that prevents such individuals from readily accessing the mental healthcare they require, even when they are legally entitled to access it. The literature identifies a need for the provision of care that is culturally responsive and easily accessible [48] to those from different migration or cultural backgrounds. However, the cultural barriers that are pertinent to the Bidoon may also be specific to their prior experiences and context and therefore further research exploring the Bidoon's cultural perspectives towards help-seeking around mental health is warranted.

## Implications for practice and policy

The findings of this study have multiple implications for clinical practice and policy, in terms of best practice for supporting the Bidoon, as well as wider implications for other stateless or displaced individuals.. In terms of clinical practice, Sandhu et al. [49] explored barriers to care in a study with mental health professionals supporting marginalised groups, including asylum seekers and refugees. The importance of outreach was highlighted as a way to build trust among marginalised communities. They identified that trust could be built through referral, peer-recommendation, and not requesting documentation. Four aspects of good practice were highlighted: outreach, facilitating access to services, communication, and disseminating information. Given concerns within the Bidoon community about how their personal information is used or shared, and reticence to approach healthcare services based on negative past experiences (in both Kuwait and the UK), these recommendations could potentially offer a framework to help overcome some of the barriers faced by Bidoon people living in the UK. Moreover, these recommendations have a wider utility in promoting access to and engagement with services for other stateless or displaced people who may struggle to access healthcare for a variety of reason.

Services must be also appropriately organised to proactively facilitate access and meet the (sometimes complex) needs of stateless and displaced people. A recent review [30] highlighted the importance of addressing practical service barriers, such as: providing financial support to travel to in-person appointments, offering flexible appointment times, ensuring access to interpreters or multilingual therapists, and the option for of digital therapy sessions. Other literature further highlights the need for professionals and services to work collaboratively to meet the holistic (including legal and social) needs of displaced people [23,50] to facilitate the individual's ability to optimally benefit from any mental health treatments provided.

Participants in this study spoke to painful experiences of discrimination from staff in statutory services, including from healthcare professionals. Experiences of discrimination is associated with physical illness, as well as feelings of guilt, anxiety, and hypervigilance [51,52]. Further, experiencing interactions that emphasise one's 'otherness' as a forcibly displaced person in healthcare settings has also been linked to a reduced ability to maintain healthy practices such as exercising, eating nutritious foods, and attending healthcare appointments [7]. Research suggests that those who have experienced discrimination in a healthcare setting find it more difficult to build a trusting relationship with professionals [53], which may explain the reluctance of one participant to seek support from mental health services following experiences of discrimination. In line with the recommendation of other studies [30], it is therefore imperative that healthcare professionals (and those working in other statutory organisations) are appropriately trained and suitably informed about socio-political contexts to be able to provide culturally sensitive care to individuals who have experienced displacement or who are managing ongoing immigration issues.

The findings also call into question the current capability of mental health services to address difficulties which are closely connected to complex socio-political issues, such as statelessness. Mental health care that is not solely located within the individual, and that encourages systemic change and community engagement may be of particular value for stateless people and communities [48]. Future research exploring such interventions and how services can be re-organised or structured to best meet the needs of stateless people is therefore urgently required.

Moreover, long-term solutions may come from not supporting individuals, but by addressing the sociopolitical and legal mechanisms that made the Bidoon and other stateless people vulnerable to mental health difficulties in the first place [42]. This should involve not only addressing local statutory service, healthcare and immigration policy issues within the UK, but also through using this understanding of the long-term negative consequences of statelessness to lobby for international policy changes to address the underlying causes of statelessness, and ultimately prevent it.

## Limitations

A key limitation is that the number of participants interviewed was very small. It is important to note, however, that this community is highly marginalised even within the UK, and it took substantial time and efforts on the part of the research team to establish connections and trust with members of the community in order for them to feel able to participate. Despite reassurances about anonymity, two people withdrew their consent due to concerns that about repercussions their family in Kuwait might face repercussions if they were identified as having participated in the study. This highlights the dilemmas the community face when making decisions to speak to professionals or researchers about their experiences. As the voices and perspectives of Bidoon individuals are rarely represented in academic literature, despite the small sample size, this study therefore represents something unique and very important.

Despite the extensive efforts on the part of the researchers, recruitment likely did not reach those who are most marginalised or who faced other structural barriers to participation. For example, only one woman participated in the study, meaning that female perspectives were under-represented. This may reflect specific cultural barriers to taking part in research [54], which needs to be considered when recruiting for any future projects with this community.

Although participants were offered detailed explanations about how their data would be stored securely and why audio recording the interviews was necessary, it is clear that despite attempts at community consultation, we did not sufficiently address the anonymity concerns of potential participants. The concern raised by the community is understandable, and reflects a 'blind-spot' on the part of the researchers who failed to appreciate the depth of fear that the community have about how information about them might be used, and made incorrect assumptions that participants would understand the strong emphasis placed on protecting anonymity and personal data within research conducted in the UK. It may have been preferable to provide more opportunity to discuss these concerns and consider alternative solutions with participants prior to the interviews being scheduled. Future research should engage with further consultation with affected communities and carefully consider how research participation can be facilitated when individuals may fear serious consequences for sharing their lived experiences and perspectives.

For interviews conducted through an Arabic interpreter, steps were taken to reduce misinterpretation by using a professional interpreter who has extensive experience in mental health settings. There was also a debriefing with the interpreter after each interview to discuss anything culturally or lingually significant that may have been lost in the translation. However, despite this, there may have been a loss or misunderstanding of the meaning imbued by the participants. Additionally, the interpreter's interpretation of meaning may have inadvertently permeated their translation, which might have biased or compromised the analysis. Nevertheless, non-English speakers were included in this study as they are likely to be amongst some of the most marginalised Bidoon people in the UK, and it was therefore important to give them the opportunity to participate in this study. In the future, measures such as back-translation, participant validation, or dual-transcription could enhance accuracy. These were not possible due to time and budget constraints of this project but could improve future IPA research with this group.

Another limitation is that due to the small sample, restricted demographic data that was collected and the nature of the interview topics, we were unable to explore in detail whether there was any difference in the experiences of people who had lived in the UK for a longer period of time or had obtained leave to remain or citizenship. We were also unable to investigate whether living in particular regions of the country (including those who might live in areas designated as city/boroughs of 'sanctuary' [55] for displaced people) had any particular impact for the participants, in

terms of experiencing more acceptance and support, or hostility and discrimination. These issues may all have been relevant to their experience of current services and perception of current safety and wellbeing, and should be further explored in any future research.

## Conclusions

This study gives voice to the experiences of a highly marginalised stateless community; it aims to raise awareness of the context for Bidoon people and how their experiences of statelessness may impact their mental health and engagement with available mental health services. It has potential implications for professionals working to support Bidoon people in the UK, or other stateless or displaced people. This includes both mental and physical health professionals working with healthcare and other statutory or third-sector organisations, as well as those working in the legal sector or Home Office. It highlights the negative experiences of navigating the UK asylum and statelessness procedures and the detrimental impact that long waits and legal uncertainty can have on the mental wellbeing of stateless individuals seeking legal protection. The findings emphasise the need for timely, accessible and tailored mental health support for Bidoon and other stateless people. Health professionals must recognise and acknowledge the strain caused by the pre- and post-migration challenges stateless people experience, and ensure that services and treatments are flexible, responsive and structured to meet their mental health and holistic needs. Further research exploring cultural attitudes and understandings of mental health, help-seeking, and intergenerational trauma arising from statelessness would provide a deeper understanding of the needs of Bidoon and how to shape effective support systems and services for them. Ultimately, these findings add to the existing literature in calling for changes to practice across healthcare and statutory services to best support the wellbeing of stateless people. This study also furthers understanding of the long-term negative mental health consequences of statelessness and highlights the needs for international policy changes to address the underlying socio-political causes of statelessness.

## Supporting information

**S1 Checklist. Inclusivity in global research questionnaire.**
(DOCX)

## Acknowledgments

We would like to sincerely thank the members of the Kuwaiti Bidoon community who bravely and generously participated in this research. We are very grateful to Sohayla Hashemi and the other community consultants (who prefer to remain anonymous) whose valuable insights were essential to the development, design and delivery of this project. We would also like to thank Dr Andreas Bjorklund and staff at Asylum Aid and the European Network on Statelessness for their support and advice on this project.

## Author contributions

**Conceptualization:** Sana Zard, Ciarán O'Driscoll, Jessie Mulcaire, Leah Holt, Francesca Brady.

**Data curation:** Sana Zard, Jessie Mulcaire, Leah Holt, Francesca Brady.

**Formal analysis:** Sana Zard.

**Investigation:** Sana Zard.

**Methodology:** Sana Zard, Ciarán O'Driscoll, Jessie Mulcaire, Leah Holt, Francesca Brady.

**Project administration:** Sana Zard, Francesca Brady.

**Supervision:** Ciarán O'Driscoll, Francesca Brady.

**Writing – original draft:** Sana Zard, Jessie Mulcaire, Leah Holt, Francesca Brady.

**Writing – review & editing:** Ciarán O'Driscoll, Sasha Menon, Francesca Brady.

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
