## [Decision Letter · Decision Letter 0]

13 Nov 2025

PMEN-D-25-00365

Statelessness and mental health experiences of Kuwaiti Bidoon people living in the UK: An Interpretative Phenomenological Analysis

PLOS Mental Health

Dear Dr. Brady,

Thank you for submitting your manuscript to PLOS Mental Health. I am really sorry for the delay. After careful consideration of the reviewer comments, we feel that your paper has merit but does not fully meet PLOS Mental Health’s publication criteria as it currently stands. Therefore, we invite you to submit a revised version of the manuscript that addresses the points raised during the review process.

Please ensure that you address all of the comments raised by reviewers, which you can find at the end of this email.

We look forward to receiving your revised manuscript.

Kind regards,

Dr Karli Montague-Cardoso

Staff Editor

PLOS Mental Health

Journal Requirements:

1. Please include a complete copy of PLOS’ questionnaire on inclusivity in global research in your revised manuscript. Our policy for research in this area aims to improve transparency in the reporting of research performed outside of researchers’ own country or community. The policy applies to researchers who have travelled to a different country to conduct research, research with Indigenous populations or their lands, and research on cultural artefacts. The questionnaire can also be requested at the journal’s discretion for any other submissions, even if these conditions are not met.  Please find more information on the policy and a link to download a blank copy of the questionnaire here: https://journals.plos.org/mentalhealth/s/best-practices-in-research-reporting. Please upload a completed version of your questionnaire as Supporting Information when you resubmit your manuscript.

2. Please ensure that your Ethics Statement is available in its entirety at the beginning of your Methods section, under a subheading 'Ethics Statement'. It must include:

1) The name(s) of the Institutional Review Board(s) or Ethics Committee(s)

2) The approval number(s), or a statement that approval was granted by the named board(s)

3) (for human participants/donors) - A statement that formal consent was obtained (must state whether verbal/written) OR the reason consent was not obtained (e.g. anonymity).

3.We note that you have indicated that there are restrictions to data sharing for this study. For studies involving human research participant data or other sensitive data, we encourage authors to share de-identified or anonymized data. However, when data cannot be publicly shared for ethical reasons, we allow authors to make their data sets available upon request. For information on unacceptable data access restrictions, please see http://journals.plos.org/plosone/s/data-availability#loc-unacceptable-data-access-restrictions.

Additional Editor Comments (if provided):

Reviewers' comments:

Reviewer's Responses to Questions

**Comments to the Author**

1. Does this manuscript meet PLOS Mental Health’s publication criteria?

Reviewer #1: Yes

Reviewer #2: Partly

2. Has the statistical analysis been performed appropriately and rigorously?

Reviewer #1: N/A

Reviewer #2: N/A

3. Have the authors made all data underlying the findings in their manuscript fully available (please refer to the Data Availability Statement at the start of the manuscript PDF file)?

Reviewer #1: No

Reviewer #2: No

4. Is the manuscript presented in an intelligible fashion and written in standard English?

Reviewer #1: Yes

Reviewer #2: Yes

Reviewer #1: This is a well structured and solid article which analyzes the experience of a narrowly-defined but interesting population: Kuwaiti bidoon in the UK. The authors dealt with significant challenges to working with a small and marginalized population, but the data they collected is rich and fascinating. The choice of IPA as a data analysis method was a good one, both because this research is clearly exploratory and because it allows for deep analysis of a relatively small data pool. The data is not being made available to other researchers, but this is an appropriate decision in the case of interview transcripts with a vulnerable population, particularly one that experiences justifiable worry about the abusive use of their data. The analysis is clear, and draws appropriate conclusions from the interview data. Overall, the article is clearly written, describes its limitations and its conclusions well, and provides interesting insight into the perspectives and experiences of its interviewees.

My primary suggestion for revision focuses on broadening the applicability of the article’s results. I should stipulate that I myself have conducted research on the bidoon, and think their situation is absolutely worthy of close consideration, but I think this article unnecessarily limits itself by focusing so narrowly on what these results can show us about the bidoon. The analysis section does link the findings to the broader literature, but I’d like to hear some reflections about what other groups might be similar to the bidoon (other stateless people? other asylum seekers? To what extent do language and culture affect the way mental health issues affect stateless people and asylum seekers?) and how mental health professionals might use this knowledge to work not only with bidoon, but with other people with shared experience. (As someone who studies politics and policy, I’d also suggest proposals for changes to the asylum process or other social supports that might help improve the situation for future bidoon asylum seekers, based on this data.

However, this article is interesting, well-structured, and uses its limited data to its fullest potential. I recommend minor revisions to ensure that its findings are best able to be applied to improve mental health care for the future.

Reviewer #2: Interesting study.

Small number of informants for one year of data collection efforts. WOuld be good to provide more clear information about the recruitment strategy.

The concluding section of the abstract should be strengthened by including clearer and more direct recommendations for both policy and practice, outlining specific actions that could improve healthcare access or support for migrants based on the study’s findings.

In the introduction or background section, it would be beneficial to add a paragraph describing the rights and entitlements of this social group in the UK, particularly regarding healthcare access, and to provide readers with essential policy context.

In the methodology, the reference to partner organisations should be clarified by naming them where possible; if anonymity is required, include a brief justification explaining why this is necessary. The inclusion criteria also need greater detail, specifying whether there was a requirement regarding the length of time participants had lived in the UK and reporting how long they had been in the country.

More information about the Arabic interpreter would strengthen transparency—describe how the interpreter was recruited, what training they had received, and the extent of their participation in the wider project. this is briefly mentioned in study limitations. other reflections re-t eh use of interpreters can be found in teh literature.

In addition, clarify what the gift voucher given to participants contained and its approximate value.

The results and analysis would benefit from acknowledging that one informant was already a UK citizen, as this likely shaped their mental health experiences differently from participants at other stages of the migration process. Including a brief reflection or analytic layer on this distinction would enrich the interpretation. It would also be valuable to link participant quotes to basic demographic or status information such as age, gender, and migration status, to help readers contextualize perspectives and strengthen analytical depth.

In the discussion, consider incorporating recent literature on the suitability and capability of UK mental health services to meet the needs of vulnerable migrants. Relevant studies include work published in the Journal of Ethnic and Migration Studies (2023) and BMJ Open (2025), (https://www.tandfonline.com/doi/full/10.1080/1369183X.2023.2181126 or https://bmjopen.bmj.com/content/15/6/e096267) both of which explore implications for interventions and service delivery. Integrating insights from these studies will help situate your findings within existing research.

Furthermore, it would be helpful to indicate where participants lived and reflect on how regional socio-economic and political environments across the UK—such as life in sanctuary cities compared with less welcoming areas—shape migrant experiences.

Finally, the paper should elaborate on the implications of these findings for service provision. Drawing on both NHS and NGO contexts, highlight how mental health support structures could be improved, what barriers persist, and how the study’s insights can inform better coordination and access for migrants. This addition will ensure the manuscript makes a strong and practical contribution to policy and service improvement debates.

**Do you want your identity to be public for this peer review?** For information about this choice, including consent withdrawal, please see our Privacy Policy

Reviewer #1: **Yes:** Emily Regan Wills

Reviewer #2: No

---

## [Editor Report · Decision Letter 1]

19 Jan 2026

Statelessness and mental health experiences of Kuwaiti Bidoon people living in the UK: An Interpretative Phenomenological Analysis

PMEN-D-25-00365R1

Dear Dr Brady,

We are pleased to inform you that your manuscript 'Statelessness and mental health experiences of Kuwaiti Bidoon people living in the UK: An Interpretative Phenomenological Analysis' has been provisionally accepted for publication in PLOS Mental Health.

Best regards,

Karli Montague-Cardoso

Staff Editor

PLOS Mental Health